# Honeycomb Biosilica in Sponges: From Understanding Principles of Unique Hierarchical Organization to Assessing Biomimetic Potential

**DOI:** 10.3390/biomimetics8020234

**Published:** 2023-06-03

**Authors:** Alona Voronkina, Eliza Romanczuk-Ruszuk, Robert E. Przekop, Pawel Lipowicz, Ewa Gabriel, Korbinian Heimler, Anika Rogoll, Carla Vogt, Milosz Frydrych, Pawel Wienclaw, Allison L. Stelling, Konstantin Tabachnick, Dmitry Tsurkan, Hermann Ehrlich

**Affiliations:** 1Pharmacy Department, National Pirogov Memorial Medical University, Vinnytsya, Pyrogov str. 56, 21018 Vinnytsia, Ukraine; voronkina@vnmu.edu.ua; 2Institute of Electronics and Sensor Materials, TU Bergakademie Freiberg, Gustav-Zeuner Str. 3, 09599 Freiberg, Germany; tsurkandd@gmail.com; 3Faculty of Mechanical Engineering, Institute of Biomedical Engineering, Bialystok University of Technology, Wiejska Str. 45C, 15-351 Bialystok, Poland; e.romanczuk@pb.edu.pl (E.R.-R.); p.lipowicz@pb.edu.pl (P.L.); 4Center for Advanced Technology, Adam Mickiewicz University in Poznań, Uniwersytetu Poznańskiego 10, 61-614 Poznan, Poland; rprzekop@amu.edu.pl (R.E.P.); ewa.gabriel@amu.edu.pl (E.G.); 5Faculty of Chemistry, Adam Mickiewicz University in Poznań, 8 Uniwersytetu Poznańskiego, 61-614 Poznan, Poland; xmifrd@gmail.com; 6Institute of Analytical Chemistry, TU Bergakademie Freiberg, Leipziger Str. 29, 09599 Freiberg, Germany; korbinian.heimler@chemie.tu-freiberg.de (K.H.); anika.rogoll@chemie.tu-freiberg.de (A.R.); carla.vogt@chemie.tu-freiberg.de (C.V.); 7Faculty of Physics, University of Warsaw, Pasteura 7, 02-093 Warsaw, Poland; pawel.wienclaw@sygnis.pl; 8Department of Chemistry and Biochemistry, The University of Texas at Dallas, 800 W Campbell Rd, Richardson, TX 75080, USA; stelling@utdallas.edu; 9International Institute of Biomineralogy GmbH, Am St.-Niclas Schacht 13, 09599 Freiberg, Germany; info@intib.eu

**Keywords:** honeycomb structure, biosilica, actin, bioinspired materials, glass sponge, *Aphrocallistes beatrix* (*A. beatrix*), 3D printing, microtomography, biomimetics

## Abstract

Structural bioinspiration in modern material science and biomimetics represents an actual trend that was originally based on the bioarchitectural diversity of invertebrate skeletons, specifically, honeycomb constructs of natural origin, which have been in humanities focus since ancient times. We conducted a study on the principles of bioarchitecture regarding the unique biosilica-based honeycomb-like skeleton of the deep-sea glass sponge *Aphrocallistes beatrix*. Experimental data show, with compelling evidence, the location of actin filaments within honeycomb-formed hierarchical siliceous walls. Principles of the unique hierarchical organization of such formations are discussed. Inspired by poriferan honeycomb biosilica, we designed diverse models, including 3D printing, using PLA-, resin-, and synthetic-glass-prepared corresponding microtomography-based 3D reconstruction.

## 1. Introduction

Bioinspiration of forms and structures observed in nature, especially in the huge diversity of skeletal constructs of fossilized and living invertebrate organisms, remains one of the driving forces of modern-material science and biomimetics. There is a current trend for the reproduction of evolutionary architecture, and exploring the possibilities of its applications in various fields of science and technology [1]. One of the classical objects produced in nature and used in bioinspired engineering [2,3] is honeycomb, built by bees (Figure 1a,b) or wasps (Figure 1c,d) [4,5,6].

Structural and mechanical properties of artificial honeycomb, such as elasticity, compressive behavior, etc., of both hexagonal (honeycomb-like) and triangular 3D structures, including hierarchical ones, are well-known and widely investigated [5,7,8,9,10,11,12]. Due to their high mechanical strength and elasticity, these structures are quite popular in modern architecture [1], engineering [3,13] and regenerative medicine [13,14,15,16], because they achieve the specified physical and mechanical properties of the final structure, while using less material. Recently, attention of researchers has been focused on effects of different structural parameters on the plateau stress and energy absorption of diverse concave hexagons with respect to thehoneycomb structure of a negative Poisson’s ratio under multi-directional impact [17]. Additionally, peculiarities of sound radiation in selected honeycomb structures are under study [18].

Honeycomb constructs in nature are mostly made of cellulose-, chitin- or wax- containing biopolymers (Figure 1) (see also [6,19,20,21]). However, such scaffolds made of biosilica, with the exception of selected structures observed indiatom shells on the nano-level, are poorly investigated [22]. In addition, honeycomb structures are found in diverse representatives of glass sponges (class Hexactinellida), which belong to the first biosilica-producing multicellular organisms on our planet (see for details [14]). Hexactinellid sponges are exclusively marine habitants and include about 550 species [23,24,25,26,27]. They can produce exoskeletons that contain up to three-meter-long needle-like spicules as well as 3D skeletal networks with more than 45 diverse complex geometries [28]. Most of them are made of amorphous silicon dioxide [29]; however, nanoinclusions of crystalline calcium carbonate have also been reported in selected species (for details see [30]). Some organic molecules such as collagen [23], chitin [31], glassin [32] and actin [28] are the templates responsible for the biosilicification processes as well as for the pattering of similar biosilica in hexactinellids. Building siliceous architecture in sponges is usually species-specific, and involves complex genomic and biochemical mechanisms. As such, the physical nature of these structures continues to be an active topic of numerous scientific discussions [14,33,34,35]. From a biological perspective, highly specialized three-dimensional bioarchitectures in glass sponges are responsible for optimal cellular attachment, for their applications as scaffolds for tissue engineering, and in the filtering of food from the water column as well as in the protection of their bodies from inorganic particles from the sea bottom under extreme conditions of deep-sea survival [35,36].

In this study, the aims are to understand the physical forces that compile to organize hierarchically structured 3D honeycomb-like siliceous skeleton of the psychrophilic (habituating in 4 °C cold waters) [37] and the less-studied hexactinellid sponge *Aphrocallistes beatrix* (Figure 2) in comparison to other glass sponges, in order to reproduce some kind of honeycomb motif using modeling, microtomography-based 3D reconstruction and 3D printing. It is recognized that the rapid development of 3D-printing technology has provided numerous opportunities for studying, reproducing, and using already-existing biological 3D structures and for the creation of new ones using a bioinspiration approach [2,13,38,39].

The glass sponge *A. beatrix* belongs to the deep-sea community [37,40]. This study shows that the nanoscale structure of *A. beatrix* skeleton is a unique combination of hexangular honeycomb-like and triangular siliceous structures in different projections. Despite the fact that the combination of triangular and hexagonal hierarchical structures are known to appear in living organisms atthe nano- and micro- (for example, DNA-origami structures [41] or diatoms frustules [1,14,22,42,43,44]) levels, their appearance at the macro-level in *Aphrocallistidae* glass sponges has not been investigated yet [45]. Until now, most attention has been focused on material-science-oriented studies of the famous *Euplectella aspergillum* glass sponge, including the modeling of its geometrically complex skeleton [46,47,48,49,50].

## 2. Materials and Methods

### 2.1. Sample Origin

*A. beatrix* (Gray, 1858) (Hexactinellida: Hexactinosida: *Aphrocallistidae*): IORAS 5/2/360 (Institute of Oceanology of Russian Academy of Sciences) RV ‘Akademik Kurchatov’—36, sta. 3724, trawl, 6 o13.2′ S54o 24.0′–26.0′ E, depth 1420–1510 m.

*Lefroyellaceramensis*: IBMRAS ID 2174 (A.V. Zhirmunsky National Scientific Center of Marine Biology) RV ‘Akademik M.A. Lavrentyev’—94, ROV ‘Comanche’, sta. 9, OdginGayotte, 37.7457072962812 N 171.104637541274 E, depth 1895 m.

### 2.2. Digital Light Microscopy

Organic-free samples of the skeleton of *A. beatrix* before and after demineralization was observed using a Keyence VHX-7000 digital optical microscope with zoom lenses VHX E20 (magnification up to 100×) and VHX E100 (magnification up to 500×) (Keyence, Osaka, Japan).

### 2.3. Scanning Electron Microscopy (SEM)

Structural analysis of the *A. beatrix* skeleton was performed using scanning electron microscopy (SEM). The samples were fixed in a sample holder and covered with carbon for 1 min using an Edwards S150B (West Sussex, UK) sputter coater. The samples were then placed in an ESEM XL 30 Philips (Amsterdam, The Netherlands) scanning electron microscope.

### 2.4. Confocal Micro X-ray Fluorescence (CMXRF) Measurements

The CMXRF analysis was performed on a modified M4 TORNADO benchtop MXRF spectrometer (Bruker Nano GmbH, Berlin, Germany), which was equipped with a 30 W micro focus Rh tube, operated at a high voltage of 50 kV and an anode current of 600 µA. Due to the modification, a second polycapillary lens was installed perpendicular to the first one in front of a 60 mm^2^ silicon drift detector (SDD). The in-depth sensitivity of the spectrometer is based on the defined three-dimensional probing volume which is formed by the confocal arrangement of those two polycapillary lenses in the excitation and the detection channels.

The samples of *A. beatrix* skeletons were fixated on an x, y and zmotorized sample stage by modeling clay. Due to the overall rather weak fluorescence energy values of the expected elements (e.g., silicon), a vacuum of 20 mbar was applied in the sample chamber. Three-dimensional measurements were conducted by moving the sample through the probing volume and subsequently receiving element distribution images (mappings) at several z positions. In total, 76 x-y element maps were measured inside a sample volume of 2.0 mm × 2.0 mm × 1.5 mm. Mappings were measured with a spatial spot distance of 20 µm in the x-y-z direction and a measuring time of 160 ms for each spot, resulting in a total set of 760,000 measuring points and spectra, respectively.

For first spectra evaluation (Si-K 1.742 keV and K-Kα 3.312 keV) and data exportation, the corresponding Bruker Software of the M4TORNADO was used. This produced two-dimensional datasets containing the location coordinates x and y, and the signal intensity (signal impulse counts at 87% ROI of the respective peak area) wasnormalized and converted into RGB color-coded images via in-house software after signal noise correction. The stacking of the individual element distribution images and converting into a three-dimensional image was performed with the Python application Mayavi. Nearest neighbor interpolation was used for volume visualization combined with shading calculations, improving the three-dimensional presentation. The signal intensities were color-coded by predefined LUT modes (planar contour grid module, Mayavi) and varying opacity values (volume module, Mayavi).

### 2.5. Isolation of Actin Filaments

The isolation of axial filaments from *A. beatrix* skeleton was performed using the “sliding drop technique” [28]. Sponge samples were first treated with 70% HNO_3_ at room temperature for 72 h order to remove possible organic impurities. Then, the selected parts of skeletons were cut and rinsed in dist. H_2_O up to pH 6.5, dried inair at room temperature and placed on Nunc™ Permanox™ (Thermo Fisher Scientific, Rochester, NY, USA) plastic microscope slides (27/75mm) in small drops of water. After water evaporation, one drop of 10% HF acid was added to demineralize biosilica in the sample. The slide was placed inside the Plexiglas Petri dish at a 10° angle and closed for HF evaporation prevention. The samples were left for 7–10 h for complete silica dissolving. The residual demineralized axial filaments were rinsed with dist. H_2_O and dried inthe air [28].

### 2.6. Phalloidin Staining

Cell Navigator^TM^ F-Actin labeling kits were used for the identification of actin within demineralized samples according to the classic approach [28]. We used *redfluorescence* iFluor^TM^ 594-Phalloidin, and *green fluorescence* iFluor^TM^ 488-Phalloidinin in this study. For preparing the working solution, 10 μL of iFluor^TM^ Phalloidin (Component A) was added to 10 mL of labeling buffer (Component B). Axial filaments isolated from the demineralized skeletons were fixed onto Nunc™ Permanox™ (Thermo Fisher Scientific) plastic microscope slides and100 µL/sample of iFluor^TM^ Phalloidin working solution was added. Samples were stained for 60 min at room temperature in a dark place. Afterwards the plates were carefully washed 5 times with distilled water to remove excess dye, dried and observed using light and fluorescent microscopy. Unused iFluor^TM^ Phalloidin stock solutions were stored at −20 °C and protected from light.

### 2.7. Fluorescent Microscopy

Fluorescent microscopy images were obtained using a Keyence BZ-9000 digital optical microscope (Keyence, Osaka, Japan) with zoom lenses CFI Plan Apo 10× and CFI Plan Apo 40×, using the GFP channel (Ex/Em = 470/525) for green-stained samples, and the TxRed channel (Ex/Em = 560/630) for red stained samples, and the bright field for comparison.

### 2.8. Tomography

Microtomographic imaging was performed on 3 sections of the studied material. The samples were scanned using a 1172 SkyScanmicroCT scanner (Bruker, Kontich, Belgium). Imaging was performed at two-pixel resolutions. In the case of the first two fragments, the pixel size was 10.51 µm, and in the last one it was 1 µm. The X-ray source operated at 59 kV/167 µA. The exposure factor of the array itself was kept at 45%. The rotation step during scanning was 0.25°. Images were saved in a 16-bit TIFF depth. The matrix resolution was 2000 × 1332 px for samples 1 and 2; sample 3 had a resolution of 4000 × 2664 px. All images were reconstructed in Nrecon software (Bruker, Kontich, Belgium). Ring Artifact Correction filtering at level 8 and Beam Hardening Correction at 64% were applied. The reconstructed sections were subjected to filtering and thresholding at the level of 100 grayscale index units. Binary images were converted to STL models using the Marching Cubes 33 algorithm.

Mesh simplification, filtering and smoothing were conducted in the MeshLab software. In the first step, filtration was applied. All generated duplicate surfaces and free objects not connected to the model were removed. Simplification was performed using the Surface Simplification Using Quadric Error Metrics algorithm [51]. It was then subjected to Laplace smoothing [52]. Again, the duplicate surfaces and zero area faces created during remeshing and smoothing were removed.

### 2.9. Three-Dimensional Printing

3D models were printed based on STL files obtained from CT reconstruction and processed as described above as well as from the computer-aided designed (CAD) models built using SolidWorks software.

The 3D models of the sponge skeleton were printed from three materials: polylactide (PLA), resin and glass.PLA models were printed using the fused deposition modeling (FDM) printing method on the Raise 3D N 2+ printer. The following printing parameters were used: layer height—0.20 mm, nozzle diameter—0.40 mm, infill density—100%, printing speed—50 mm/min. A PLA filament with a diameter of 1.75 mm was used. The Formlabs Form 2 printer (Formlabs GMBH, Berlin, Germany) with an accuracy of 50 µm and the stereolythography (SLA) method wereused to print the resin model.

Glass sponge models were produced with the SYGLASS_01 technology developed by SYGNIS S.A. The SYGLASS_01method uses a low-temperature glass 3D printer. To create complex geometrical shapes such as a sponge skeleton, glass is melted in the crucible mounted on the 3-axis gantry inside of the chamber. Subsequently, pressure is applied inside sealed crucible, and the kinematic system follows a preprogrammed shape, extruding melted glass in the desired positions. To print the sponge skeleton models, custom soft types of glass doped with lead and phosphate were used [53]. The following printing parameters were used: layer thickness 0.6 mm, layer width 0.8 mm.

### 2.10. Cowering of 3D-Printed Sponge Reconstruction with Diatomit

3D models of the sponge reconstruction were printed from the PLA filament with the addition of diatomaceous earth (Diatomit from Perma Guard, Inc., Bountiful, UT, USA). The filament was obtained by the method described in the paper [54]. A filament with a diameter of 1.75 mm was used. The following printing parameters were used: layer height—0.18 mm, nozzle diameter—0.40 mm, infill density—100%, printing speed—80 mm/min, extruder temperature:220 °C.

## 3. Results

The *A. beatrix* skeleton (Figure 3) has a unique hexactinellid sponge honeycomb-like structure (Figure 3 and Figure 4). It has a kind of hierarchical organization, because the walls of the hexangular tubular pores (Figure 4a–c) are formed with the hexagonal silicified structures of diverse sizes (Figure 3 and Figure 4d–f). This combination results in the forming of a triangular silicified matrix in one projection and hexangular honeycomb-like matrix in the other projection. The size of the honeycombs in *A. beatrix* may vary depending on the location of the structure; however, the hexangular pores shape remains to be constant (Figure 3) (see also Appendix A).

SEM images of the *A. beatrix* allow us to detail the hierarchy of the sponge skeleton, except hexangular pores, (Figure 5a,b) with walls formed from a mesh of triangles (Figure 5c,d) resulting from the fusion and subsequent biosilification of hexactinellid spicules; one can observe the presence of an internal axial channel with a proteinaceous axial filament inside the glass structures (Figure 5e) and the specific structure of colloidal biosilica, similar to the previously described one in other species of glass-sponges [13,39,55].

Both the digital (Figure 3 and Figure 4) and scanning-electron-microscopy (Figure 5) imagery suggest that the skeleton of the *A. beatrix* glass sponge is highly optimized for water filtering in diverse directions. This natural biosilica-based filter, being a mechanically robust construct, is also able to selectively pass microparticles of a certain size and shape. In living sponge, corresponding specific cells are localized on the surface of such highly porous skeleton. They are responsible for creating a flow of water with feed through abiological “sieve”. However, as recently reported for the closely related *A. vastus* glass sponge, these species “typically go into temporary apparent pumping arrest whereby they cease filtering particles from surrounding water to prevent particle obstruction” [56].

**Figure 5 biomimetics-08-00234-f005:**
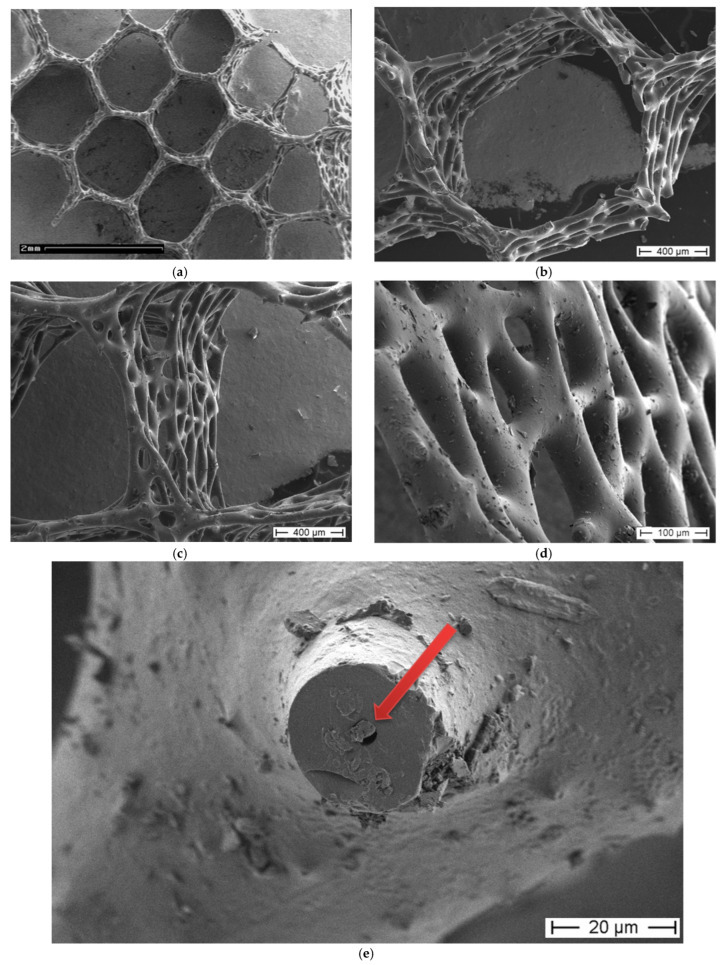
SEM images of the *A. beatrix* sponge skeleton sample: (**a**,**b**) hexangular honeycomb-like macroporous structure; (**c**,**d**) the glassy wall of the pore with the triangular structure; (**e**) axial filament (arrow) in the axial channel inside the siliceous skeleton. (See also Figure 6).

In recent examinations, Confocal Micro X-ray Fluorescence (CMXRF) has already been successfully used for element analysis of spongin-based skeletal fibers of marine demosponges *Hippospongia communis* [57] as well as chitin-based demosponges *Ianthella basta* [58]. This method ofanalysis of the siliceous sponge structures of *A. beatrix* skeleton was chosen in this study.

CMXRF analysis was executed in the region of a hexangular pore of the *A. beatrix* sample (Figure 7a). The fluorescence lines of Si-K and K-Kα yielded sufficient signal intensity for the three-dimensional rendering of the element’s micro scalic distribution within the siliceous scaffolds (Figure 7b,c) with an approximate depth resolution of 70 µm for Si-K and 60 µm for K-Kα radiation. The 3D renderings for both elements illustrate the tubular architecture of the analyzed canal in the glass sponge. By adding two-dimensional contours, the structural architecture of the glass sponge wall (Figure 7d) and honeycomb-like tube are highlighted (Figure 7e,f). Those contours visualize the Si-K signal distribution at a selected planar cross-section of the 3D data set.

Despite the low spatial resolution measuring the fluorescence lines of lighter elements, the x-z contour of the silicon signal exhibits the mesh network structure by the heterogeneous distribution of the silicon intensity due to the siliceous framework, which forms the sponge’s walls. In addition, the x-y contours demonstrate the hexagonal geometry of the tubular scaffold at different analysis depths within the glass sponge under study (Figure 7f). However, it should be mentioned that due to the low signal intensity of the K-Kα and the low fluorescence energy of the Si-K radiation, the detected signals are strongly dependent on absorption effects in the sample matrix. This can be observed in the distribution images of silicon and potassium measured at greater depths (Figure 7b,c) by decreasing signal intensities and eventual signal erasure due to applied noise correction. Nevertheless, the total analysis depth of about 1.5mm demonstrates the potential of the depth-sensitive benchtop CMXRF for non-destructive 3D analysis of element distribution within the *A. beatrix* glass sponge.

Digital microscopy of the organic-free *A. beatrix* skeletons samples before and after the procedure of partial demineralization (desilicification) have confirmed that the hexangular honeycomb-like structure of the biosilica are formed by the axial filaments, located inside the biological glass structures (Figure 8a,b).

Similar organic matrices have been recently isolated from complex skeletal frameworks of such glass sponges as *Farrea occa*, *Euplectella aspergillum*, *Malacosaccus* sp., *Walteria flemmingi* as well as from *A. beatrix* and identified as F-actin (see for details [28]). Additionally, nano-bridges typical for F-actin fibers [28,59,60,61] have been observed on SEM images of *A. beatrix* samples under study, where partially demineralized axial filaments show their nanofibrillar organization (Figure 6).

Finally, demineralized samples of the *A. beatrix* skeletons were stained with two different Phalloidin dyes, which are recognized [28,62] as highly specific for identification of F-actin (Figure 9 and Figure 10).

Stained samples of demineralized axial filaments (both with actin red- and green-fluorescence labeling kits) have shown characteristic fluorescence, confirming the presence of actin in them (Figure 9 and Figure 10). Furthermore, the fluorescent organic material in observed samples keeps the same size and shape as the honeycomb-like structure of the natural *A. beatrix* sponge skeleton studied here (Figure 9 and Figure 10). It is clear that at higher magnifications characteristic structures of actin filaments of corresponding axial filaments are presented in the form of bundles (Figure 10b,d). Previously, actin filaments have been already identified, although only within triangular structures in the glassy skeletal walls of *A. beatrix* [28]. Thus, it can be suggested that in the case of highly sophisticated siliceous skeleton of *A. beatrix* F-actin remains a crucial player in the pattering of this uniquely structured biosilica.

As it was shown recently by Sala et al. [63], both triangular and hexangular artificial 3D matrices show extremely different mechanical properties in different projections in contrast to symmetrical rectangular ones. However, the vertical-horizontal combination of these structures in the sponge’s skeleton could be the way for the necessary balance of the mechanical strength that was optimized by the *A. beatrix* glass sponge during evolution. So, it was decided in this study to reproduce the geometry of the glass sponge skeleton using the 3D printing method.

The schematic view of the 3D-modelling process based on the hierarchical structure of the *A. beatrix* sponge skeleton is shown at Figure 11.

After scanning of the selected *A. beatrix* sponge samples with microtomographic scanner the resulting STL model was very complex including the mesh that contained about 120 million triangles (Figure 12). Additionally, the reconstruction was characterized by high anisotropy. A multi-layered structure connected by bridges became visible. It is worth noting that the model was heterogeneous with different wall thicknesses. So, mesh filtering, simplification and smoothing were performed, and the model was simplified to 10 million faces to make it possible to process it in software used in 3D printing (Figure 11).

Instead of a 3D CAD model showing a full replica of the *A. beatrix* skeleton structure with equal shapes, simplified models with an ideal cylindrical shape scaled to the real size of the sponge were created (Figure 13 and Figure 14).

The models assumed that there was a hexagonal structure resembling a honeycomb on the walls, with equal spacing between the structures (Figure 13a). In addition, the models show that the walls of the honeycomb-like structures of sponge skeleton are multi-layered, with a hierarchical structure; therefore, triangle-shaped spaces were designed in the cross-section of the walls (Figure 13b,c). The outer diameter of the cylindrical 3D model was 20mm. The geometry of honeycomb cells, the wall’s structure, and the dimensions of the model are shown on Figure 14.

The simplest cylindrical model with honeycomb-like structure of the walls was printed from PLA (Figure 15). Its diameter was 5 cm. Digital microscopic analysis of the printed model showed some thin PLA filaments visible in the pores (Figure 15d,e). This probably was caused by being too small for the accurate printing size of the model walls and the hexagonal spaces in it.

More complex models were printed from different materials (PLA, resin) using different 3D printing methods (FDM, SLA) reproducing the simplified microtomography-based CAD model (Figure 16). The size of the models was 20 cm in diameter—much bigger in comparison to the natural sponge skeleton (about 2 cm in diameter)—so obtained samples were better at visualizing the structure of the primary object. However, the irregular shape and arrangement of pores was noticed. The models did not have a layered structure of walls, which was due to the simplification.

## 4. Discussion

The current running theory states that the conserved biomaterial morphology of sponges suggests the ancient role that the fundamental structural protein actin has played for more than 500 million years [28]. It has been proposed that F-actin was localized in an ancestral, intracellular siliceous construct, and as spiculogenesis moved to extracellular spaces, the actin continued to play its pattern-forming role, leading to a release on the size constraint of fibrous actin complexes [28]. Structural features such as bifurcation, dichotomic growth, and branching ability of actin at the molecular and nanofibrillar level are well recognized [64,65,66,67] and principally differentiate this structural protein from collagens, keratins, elastins, and biosilica-templating proteins such as glassin or silicateins [32,33,34]. Additionally, honeycomb-like actin filaments have been already reported to be associated with chloroplasts in ferns, mosses and seed plants [68,69].

Thus, according to this hypothesis, the branching capability of the actin filament bundle was an exaptation that led to the diverse, sometimes very sophisticated composite-based skeletal forms at the macro level in glass sponges. In other words, the “epitaxy” of biosilica structures well described in a network connectivity (i.e., monaxons, triaxons, tetraxons) [28], and superficial ornamentation of fused skeletons (Figure 3, Figure 4, Figure 5 and Figure 8) is due to the presence, branching and growth of actin filaments (Figure 6 and Figure 10). However, the mechanisms of gene-related regulation of this phenomenon remain unknown despite its fundamental role.

Regarding the structural biology of glass sponges, the following can be suggested. The honeycomb-like skeleton construction is a very peculiar feature, which characterizes a unique family of sponges—*Aphrocallistidae*—with only two genera: *Aphrocallistes* and *Heterochone* [70]. No visible dictyonal strands, structures typical for many genera which have dictyonal skeletons and which characterize their primary skeletons [71], may be observed in both *Aphrocallistes* and *Heterochone*. The wall of the honeycomb unit is constructed from a thin, likely single layer of hexactinic spicules which are fused to their neighbors at points of mutual contacts. This type of the skeleton, when it is situated circumcircle (constructs the wall of vase-like or tubular body), is called aulocalycoid [71]. Loose (not fused) spicules of different types are also present in living representatives of the family, which together with the rigid framework provide the support of the sponge ‘living body’ and make the base of the circumcircle oriented to the dermal and atrial skeletons. Recent laboratory investigations on the spicule formation of *A. beatrix* [72] did not provide information on the honeycombs growth and the only possibility now is searching for any tracks of this reconstruction among other hexactinellids.

The most primitive ancestor of hexactinellid sponges had a tubular, cap- or funnel-like body shape [71] and further evolution may be connected to body enlargement followed by wall thickening, which demands the development of canals. The case of the canals (caves between the honeycomb-structured skeleton) in *Aphrocallistidae* has a specific term—diarhyses. The appearance of unique diarhyses and a corresponding honeycomb-like pattern of the rigid skeleton has never been discussed before.

An important genus in this reconstruction (after a primitive tubular of funnel-like *Chonelasma*—family Euretidae) is *Lefroyella* (Euretidae) but its body form interpretation is completely wrong in recent publications [70,73]: “*Euretidae with funnel-like, erect body form … constructed of a layer of small-caliber, longitudinal tubes forming longitudinal ridges separated by deep longitudinal grooves on atrial lining*…”. As it is seen from the cleaned rigid skeletons of representatives of *Lefroyella ceramensis* [73,74] they have funnel-like body with vertically directed ridges on the atrial (inner) side of the funnel. This fact simultaneously provides the transfer of the genus from the subfamily Euretinae (“*Body composed of dichotomously branching tubes*” [75]) to Chonelasmatinae (“*Body tubular, funnel-form or blade-form but without dichotomous branching*” [75]). An important fact is that the body of *Lefroyella* is funnel-like (not tubular) which leads to the dichotomous branching of the ridges at the atrial side (Figure 17 and Figure 18a) and to appearance of intercalary ridges [74], which follow the increase in the body diameter upwards. Small bridges between the neighboring ridges were also described [73].

The next step in the formation of a honeycomb-like construction is the fusion of corresponding ridges. The fusion of a framework skeleton in hexactinellids is observed during the dichotomous branching of tubular bodies [71], when two opposite sides of the tube of the upper part of the body fuse to each other and two other sides form two tubes. The line of fusion is called the carina. To imagine this process with ridges in *Lefroyella*-like sponges together with the development of framework on the atrial side with some openings from dermal and atrial surface, we will deal with representatives of the family *Tretodictyidae* with schizorhysal type of channelization. The family *Tretodictyidae* highly likely has a polyphyletic origin due to different origins of their specific channelizations. At least one type is observed in *Psilocalyx* [71], when channelization is presented by excavated dermal space, other type described above is likely observed in such genera as *Hexactinella*, *Tretodictyum*, where the schizorhyses are atrially lined surfaces.

If a hypothetical sponge skeleton receives more order through: periodicity in dichotomous branching and fusion of ridges, constant growth of the ridge, and disappearance of dermal and atrial frameworks (fused skeletons), the regular honeycomb construction observed in *Aphrocallistidae* with diarhyses channelization (Figure 18b) can be obtained. Thus the most likely origin of *Aphrocallistidae* is that from *Euretidae* (*Lefroyella*-like sponges), through *Tretodictyidae* (*Hexactinella* or *Tretodictyum*-like representatives). At least the version suggested above explains the very difficult fact in the construction of the skeletons of hexactinellids observed in *Aphrocallistidae*: the presence of the fused skeleton is situated in the radial plane, unlike other hexactinellids with rigid skeletons where the main framework is situated in the tangential plane (about circumference). Which leads to the statement: “*The biological reason for the presence of actin filaments assembly within such higher-ordered biological materials as skeletal 3D networks can be based on the collective behavior of systems composed of many thousands of actin filaments to mediate increase inthe surfaces of siliceous constructs which are vital for maintaining the integrity of the sponge and its survival in aquatic environmental conditions being feed filtering and sessile organism*” [28].

Without a doubt, honeycomb structures in nature have a different purpose than in material science and engineering. The cellular structural motif in the skeletons of organisms implies the benefits of biomaterial saving, provided that a level of strength is maintained that is almost ideal for the survival and functioning of the organism. Glass sponges in general, and *A. beatrix*, as one of the sponges with a complex, highly regular and structured skeletal formations, represent a typical example of mineralized, natural, composite materials with high biomimetic potential known as sources for 3D printing, computation, and testing [76]. Consequently, from a human view, diverse natural honeycomb structures, including those described in this study, represent the main driving force in the bioinspired design of energy absorbing, lightweight cellular composites. Nowadays, this topic remains to be excellently analyzed in several review papers (i.e., [12,77,78,79]).

## 5. Outlook

The existence of natural hierarchically structured honeycomb constructs made of biosilica as in the case of *A. beatrix* glass sponge, can stimulate attempts to produce similar bio-designed scaffolds from synthetic glass or fiberglass. However, in contrast to biosilica that is synthesized by diatoms, or sponges in the range of temperatures between −20 °C and 98 °C, and −1.95 °C and 25 °C [80] respectively, synthetic glasses have been produced in the temperature range from 500 °C to 1700 °C [81]. Another challenging task remaining is the development of structures similar to these glass-sponge 3D constructs with the same dimensions using existing technologies. Low density and promising mechanical properties of such constructions can allow their use in various fields of engineering, where classical honeycomb structures are widely used, including, but not limited to, the design of commercial products, civil engineering and even the aerospace industry.

In the models obtained from glass using so called SYGLASS_01 technology (Figure 19), another model simulating porosity as described above has been used. The pore structure was assumed to be gyroid, which resulted from the properties of the technological process of glass-printing itself. This structure is similar to the designed honeycomb mode; however, it is still not of such quality as the PLA-based models, and is far from a near-perfect naturally synthesized *A. beatrix* glassy honeycomb-based skeleton.

Additionally, this study attempts to modify PLA-based 3D-printed honeycomb models (Figure 20a) with biosilica in the form of commercial diatomite. Diatomite layers have been well integrated into PLA matrix and remain to be present after 5 h of sonication (Figure 20b,c). The next steps will be dedicated to the optimization of this simple technology to obtain 3D constructs with high BET due to the nano- and microporosity of diatoms frustules. Additionally, studies into perspectives of practical applications of such biomimetic hierarchical honeycomb constructs on macrolevel in catalysis and waste waters remediation are planned due to the significant specific surface area of the described bio-designed 3D structures.

Special attention should be focused on application of the described hierarchical honeycomb-based glassy constructs in biomedicine and bioengineering (i.e., bone remodeling, creation of prostheses) as well as in the development of artificial organs [82], taking into account the fact that such bio-inspired biomimetic structures, in addition to the well-known advantages oftheir physical and mechanical properties, have chemical inertness, which opens up significant prospects for their in vivo use.

In this work, preliminary modeling of sponge models in the CAD program was performed (Figure 14). The next step will be to print the bio-designed models and study their mechanical properties. The proposed models will be modified and improved, e.g., the distance between the honeycombs, the change in the regularity of the shape, the size of the honeycomb, and the cross-sectional shape of the model walls. Special attention will be paid to the modeling of the triangular pores of the glass honeycomb wall and the study of the influence of their area and structural geometry on the general mechanical properties of hierarchically structured models. In addition, a comprehensive examination of the glass sponges’ skeletons using microtomography, structure reconstruction, and 3D printing from various materials will be carried out.

## 6. Conclusions

From a bioinspirational view, the construction of the *A. beatrix* siliceous skeleton is quite interesting for the design and creation of bioinspired 3D objects of diverse dimensions. Despite the fact that the complexly organized skeletal structures of glass sponges do not ideally correspond in terms of human understanding of symmetry, the very existence of such formations in nature makes us think about the existence of some “engineering genes” that can regulate the biosynthesis of silicates in very special conditions of the marine environment of sponges. Confirmation of the presence of nanofibrillar actin within highly organized structural biosilica in this sponge additionally supports the hypothesis of the fundamental role of this biopolymer in the pattering of micro-and nano-architecture of such kinds of biological glass. Inspired by poriferan honeycomb biosilica, in this study, diverse models including that through 3D printing using PLA, resin, and synthetic glass have been designed; in addition, their corresponding microtomography-based 3D reconstructions have been prepared for the first time. Future work will seek to address the challenging task of using *A. beatrix*-like biomimetic 3D models for patterns for 3D-printing applications by incorporating other molecular species into the scaffold, improving the existing models geometry and the study of their mechanical properties to enable their use in various fields of science and engineering.

## Figures and Tables

**Figure 1 biomimetics-08-00234-f001:**
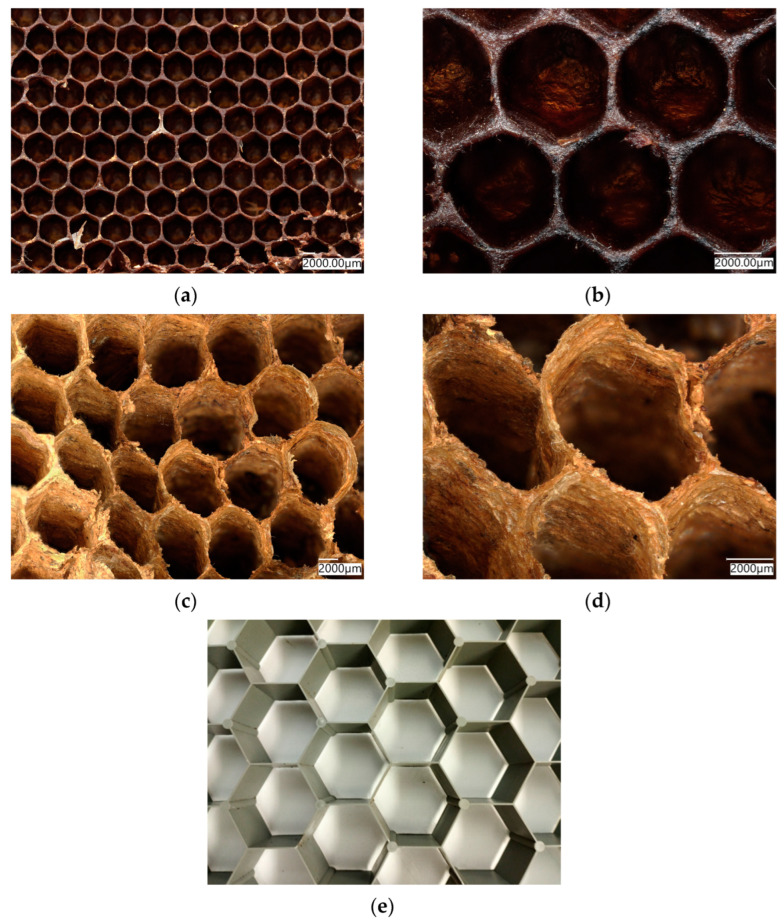
Natural biopolymer-based honeycomb structures produced by bees (**a**,**b**), or produced by wasps (**c**,**d**) have been reproduced numerous times through artificial honeycomb-like structures in larger dimensions, such asthis plastic network with 5 cmlarge segments in diameter (**e**).

**Figure 2 biomimetics-08-00234-f002:**
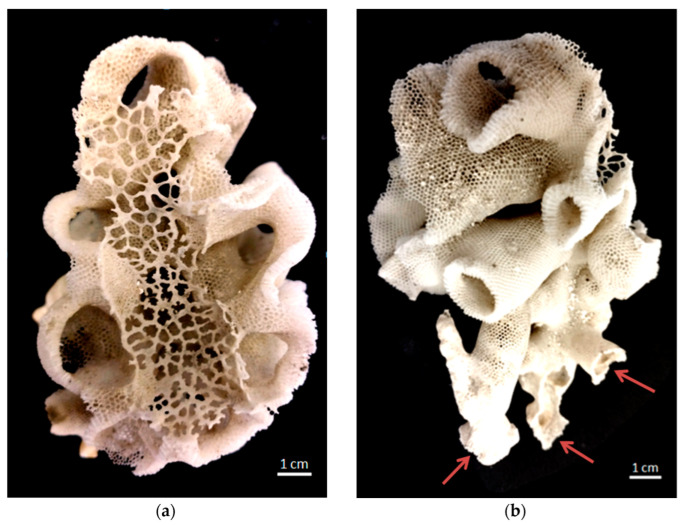
Photograph of honeycomb-like siliceous skeleton of the glass sponge *Aphrocallistesbeatrix*: (**a**) view from above: sieve-plate covers the atrial cavity; (**b**) view from the side: in the lower part several holdfast-like points (arrows) of attachment to rocky substrate are seen.

**Figure 3 biomimetics-08-00234-f003:**
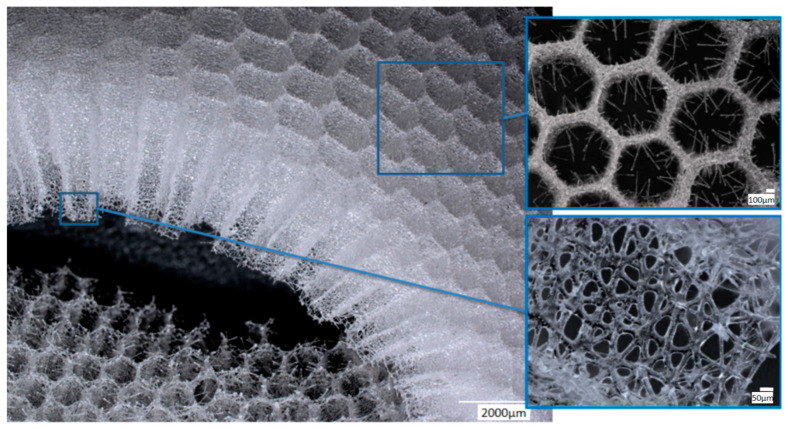
Combination of both hexangular and triangular geometries in hierarchical honeycomb-like structures of the *A. beatrix* glassy skeleton is well visible on digital light microscopy images represented here. This glass sponge constructs unique scaffolds that consist of regularly arrayed hexagonal cylinders.

**Figure 4 biomimetics-08-00234-f004:**
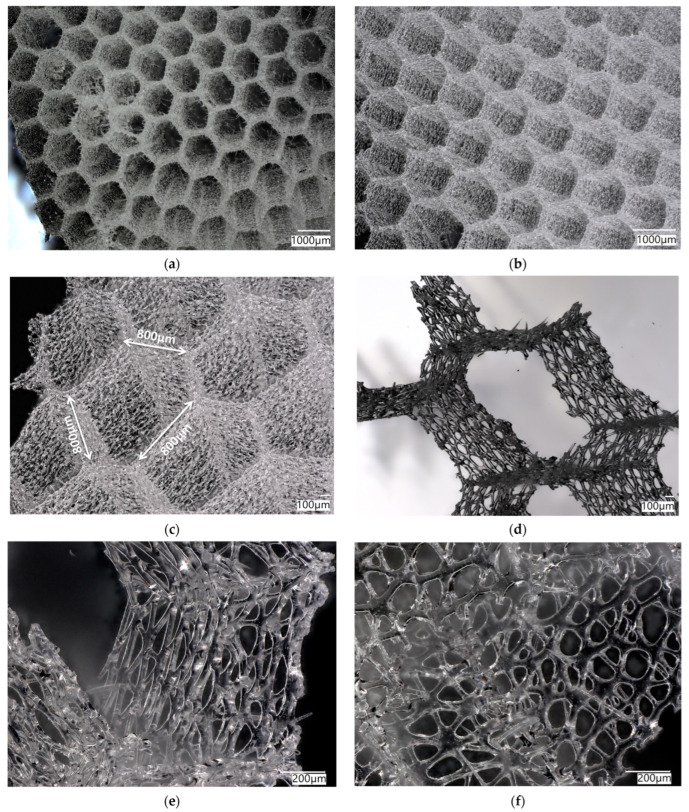
Digital microscopy of the honeycomb structures in the *A. beatrix* skeleton observed under diverse angles and magnification (**a**–**d**). Additionally, the honeycomb-like and triangular structural motifs are to be found in the glassy walls of the sponge skeleton (**d**–**f**). Corresponding measurements are represented in the Appendix A).

**Figure 6 biomimetics-08-00234-f006:**
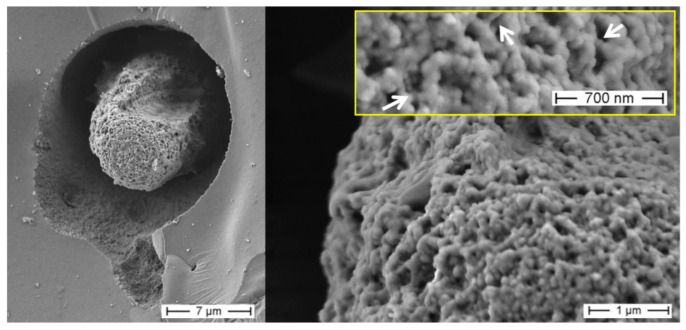
SEM imagery of the partially demineralized axial filament from a selected sample of *A. beatrix* glass sponge. This organic structure is made of a network of twisted nanofibers connectedwith each other through nano-bridges (arrows), afeature that between structural proteins is characteristic only for actin filaments.

**Figure 7 biomimetics-08-00234-f007:**
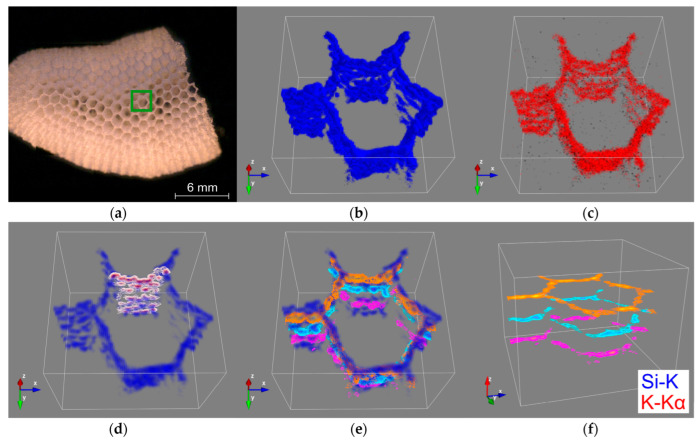
CMXRF of *A. beatrix* honeycomb skeleton: mosaic image of the analyzed sample with the marked analysis area of a hexangular porein green (**a**); 3D distribution images of silicon (blue) (**b**) and potassium (red) (**c**) within a volume of 2.0 × 2.0 × 1.5 mm; transparent silicon distribution image (scalar opacity unit distance of 0.5) with visualized x-z contours (white to dark red) of the hexagonal cell wall (**d**) and three x-y contours at different z positions (magenta, cyan, orange) with (**e**) and without (**f**) silicon visualization.

**Figure 8 biomimetics-08-00234-f008:**
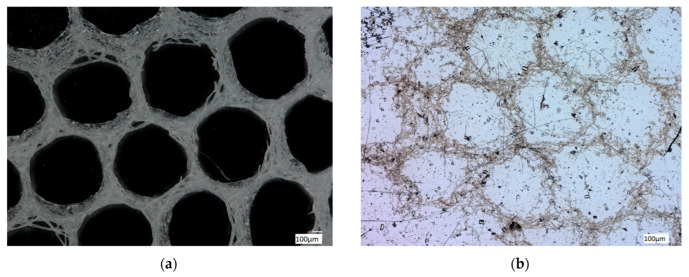
Digital light microscopy images of the organic-free skeleton fragment of *A. beatrix* sponge (**a**) and the same after demineralization with 10% HF (**b**). The organic scaffold (**b**) resembles the size and shape of the siliceous exoskeleton of this sponge (**a**) very well. See also Figure 9 and Figure 10.

**Figure 9 biomimetics-08-00234-f009:**
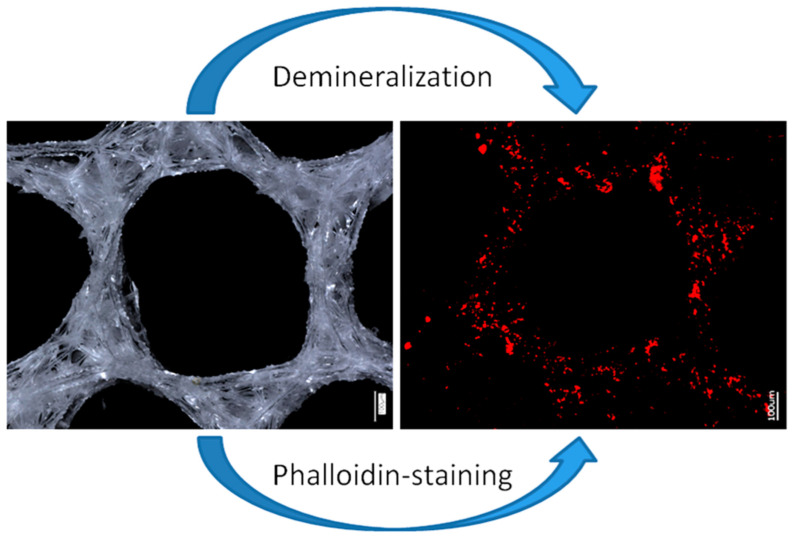
Principal schematic view of isolation of an organic matrix from glass sponge skeleton. Digital (**left**) and fluorescence (**right**) microscopy of the honeycomb-like structures of the *A. beatrix* skeleton before and after demineralization as well as direct visualization of actin filaments with phalloidin staining.

**Figure 10 biomimetics-08-00234-f010:**
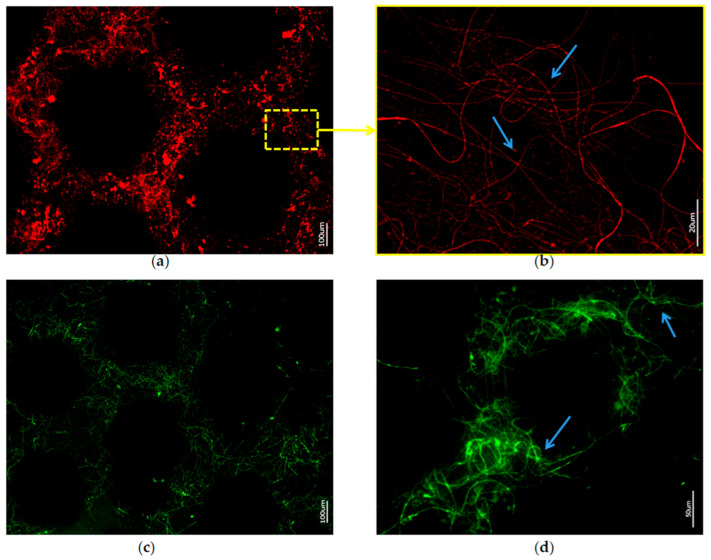
Fluorescence microscopy of axial filaments in with HF-desilicified *A. beatrix* skeletons: (**a**,**b**) stained with 594-Phalloidin; (**c**,**d**) stained with 488-Phalloidin. Arrows show the individual actin filaments agglomerated in bundles after desilicification.

**Figure 11 biomimetics-08-00234-f011:**
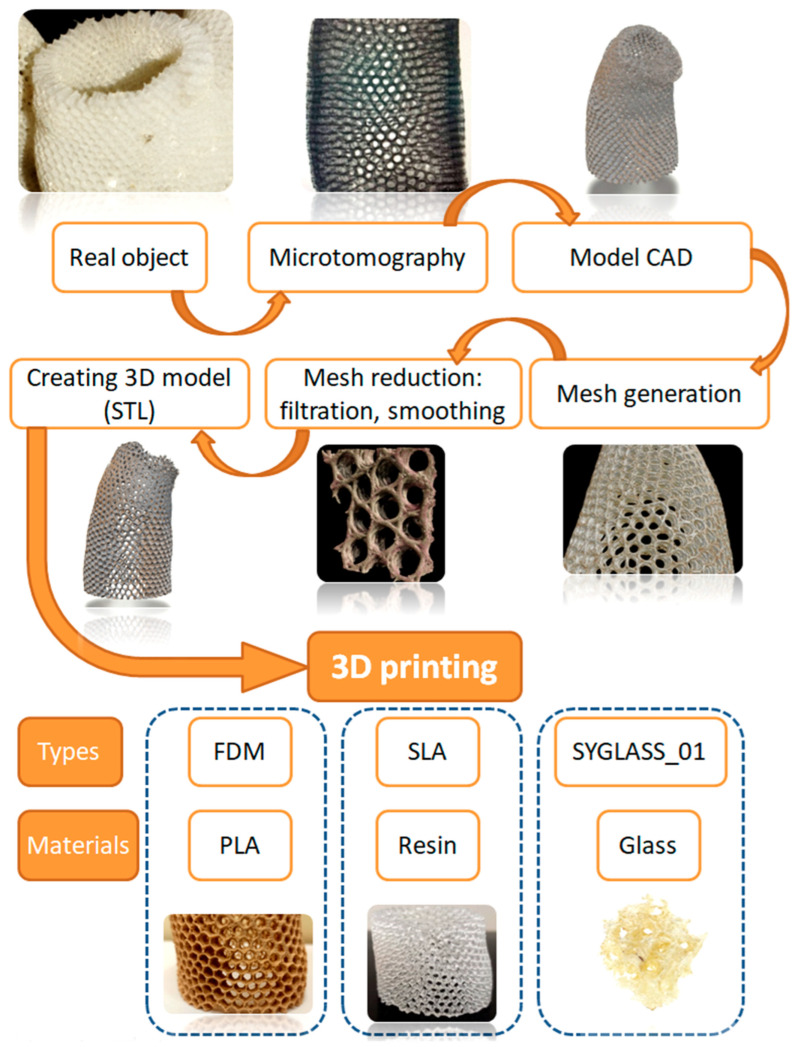
The schematic view of the 3D-printing process of *A. beatrix* skeleton 3D models.

**Figure 12 biomimetics-08-00234-f012:**
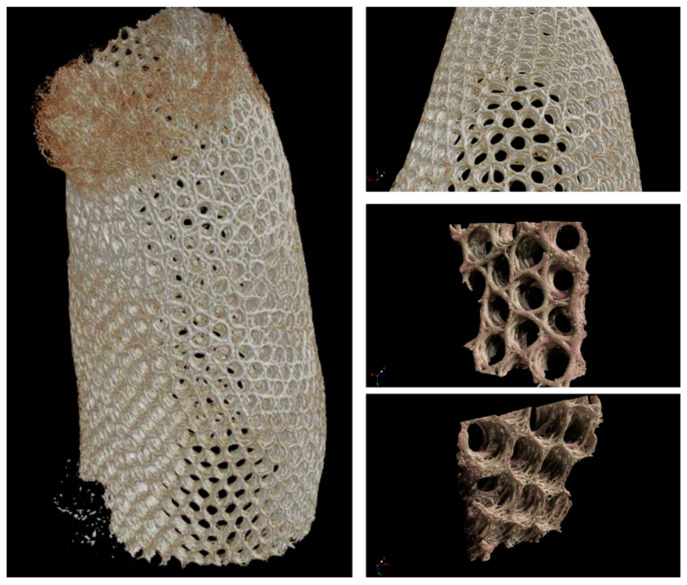
Microtomography-based3D reconstruction of *A. beatrix* glass sponge skeleton hierarchical structure.

**Figure 13 biomimetics-08-00234-f013:**
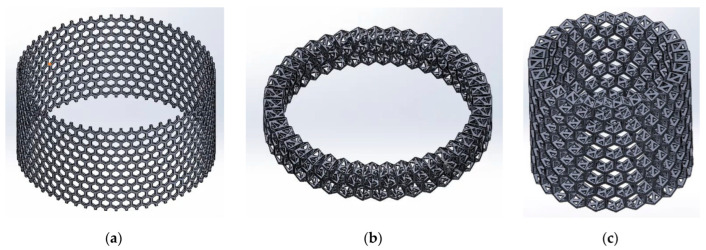
The schematic view of the 3D model of *A. beatrix*: (**a**) cylindrical flat honeycomb-like structure; (**b**,**c**) hierarchical tubular honeycombs with triangular holes in the walls.

**Figure 14 biomimetics-08-00234-f014:**
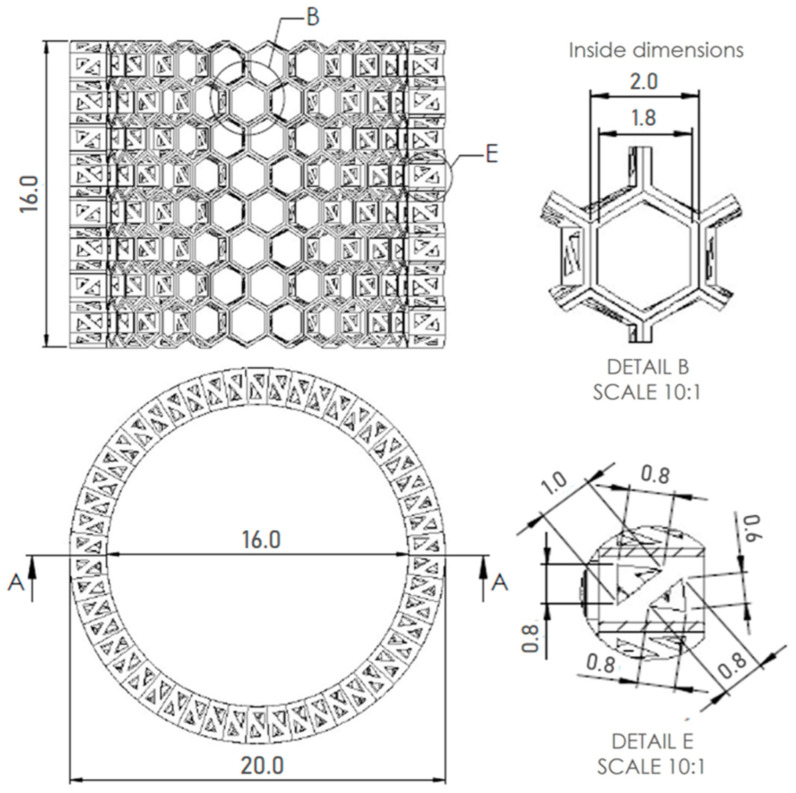
A. 3D CAD model of *A. beatrix* glass sponge skeleton with equal shapes; technical drawing with dimensions of the model, honeycomb, triangles in cross-section.

**Figure 15 biomimetics-08-00234-f015:**
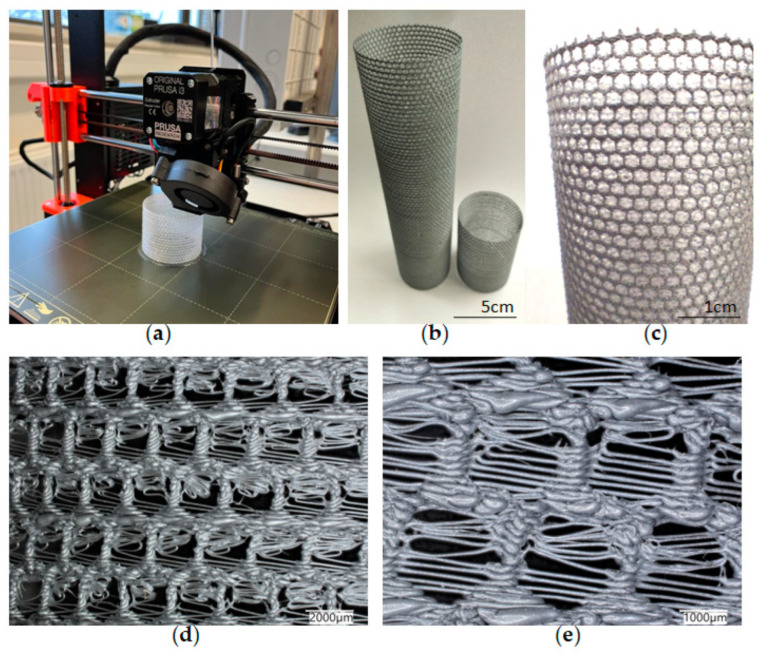
(**a**) The process of 3D printing of the model of *A. beatrix* skeleton from PLA using FDM technology; (**b**,**c**) general view of the resulting CAD-based simplest cylindrical model; and (**d**,**e**) digital microscopy of a designed model wall.

**Figure 16 biomimetics-08-00234-f016:**
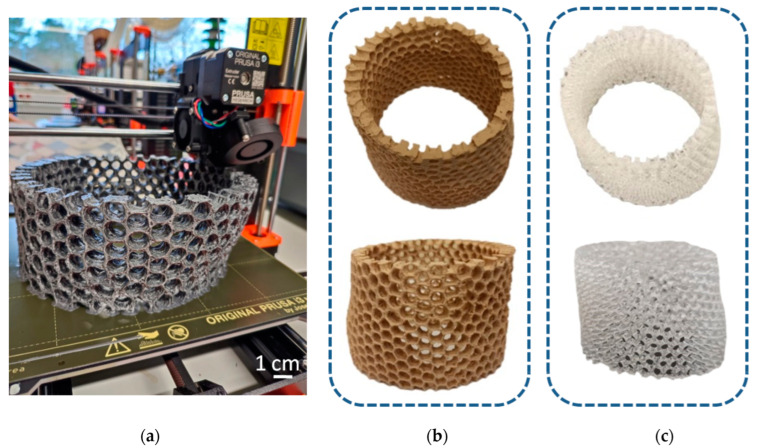
(**a**) The process of 3D printing in the reconstruction of *A. beatrix* skeleton; the resulting models made from: (**b**) PLA and (**c**) resin.

**Figure 17 biomimetics-08-00234-f017:**
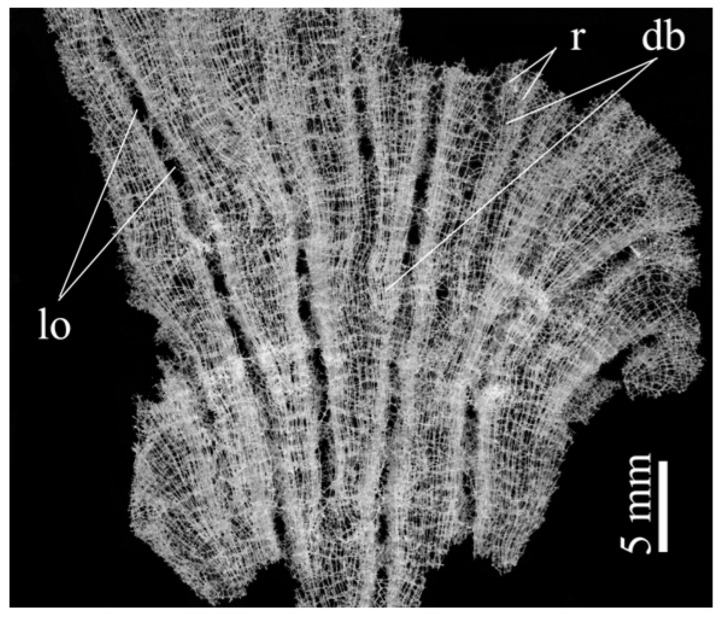
Digital microscopy image of wall fragment with the dictyonal skeleton of *Lefroyella ceramensis* from the Emperors Mountain Chain. Lo—lareal oscula; r—ridges; db—dichotomous branching of ridges.

**Figure 18 biomimetics-08-00234-f018:**
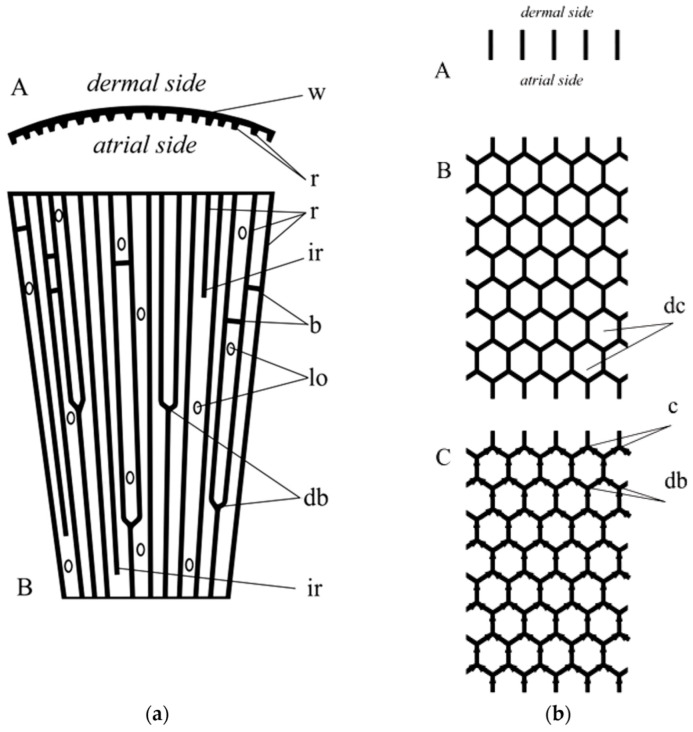
(**a**) Scheme of the wall fragment of *Lefroyella*. A: horizontal section; w—wall; r—ridges. B: longitudinal section; r—ridges; ir—intercalary ridges; b—bridgees; lo—lateral oscula; db—dichotomous branching of ridges. (**b**) Scheme of the wall fragment of *Aphrocallistes*. A: horizontal section. B: longitudinal section; dc—honey-comb unit. C: longitudinal section, arrows show the suggested direction of the growth of the dictyonal skeleton; c—carina (line of fusion; db—dichotomous branching). It is of note here that dichotomous branching of actin filaments remains to be characteristic for this structural protein [67].

**Figure 19 biomimetics-08-00234-f019:**
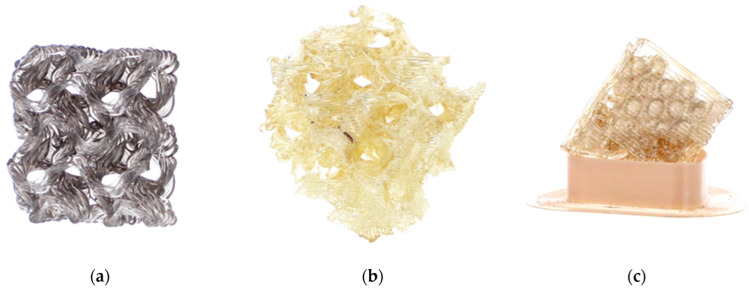
The 3D model of *A. beatrix* made from synthetic glass: (**a**) flat gyroid (dimensions 20 mm × 20 mm × 6 mm), (**b**) cubic gyroid (30 mm × 30 mm × 30 mm) and (**c**) flat structure relatively similar to the sponge skeleton samples.

**Figure 20 biomimetics-08-00234-f020:**
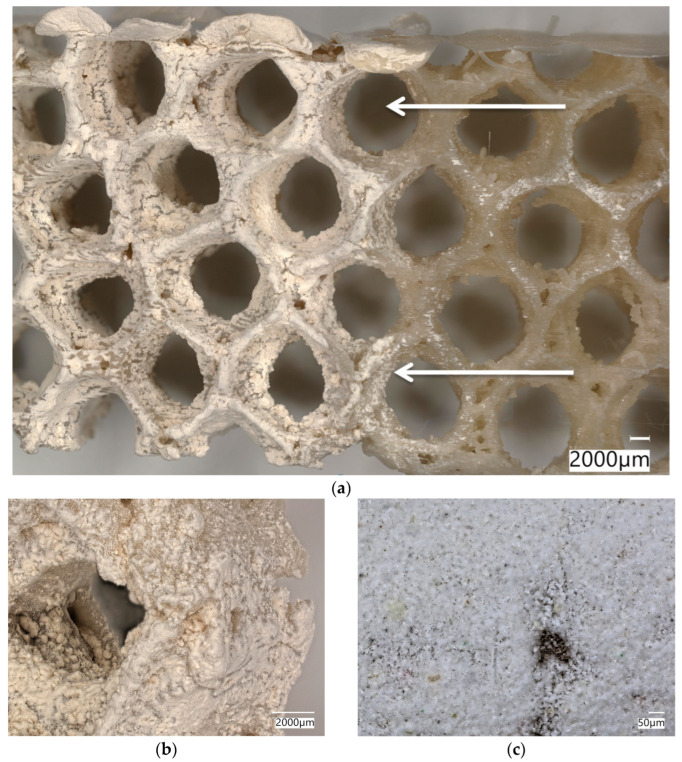
Digital microscopic images of the 3D-reconstruction-based model (**a**) of *A. beatrix* glass sponge skeleton made of PLA covered with the diatomite layer (arrows). Diatomite remains to be strongly attached to the surface of the PLA-based construct even after sonication (**b**,**c**).

## Data Availability

Not applicable.

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
