# Peer review of "Honeycomb Biosilica in Sponges: From Understanding Principles of Unique Hierarchical Organization to Assessing Biomimetic Potential"

_biomimetics, 2023, doi:10.3390/biomimetics8020234_

Round 1

Reviewer 1 Report

1. Results are mostly described, their detail is not provided, give the reasons and their influence depending on the applied parameters.

2. Please add quantitative analysis.

3. Please state the conclusions of this study.

4. What are the applications of this study. Authors are suggested to provide some technical expressions on the application of the study.

5. The author mentions the porous structure in the introduction. But, the introduction secti-on is very briefly organized. The introduction section should be improved using the paper-son the subject of this work in such as: doi:10.3390/ma16083262; doi:10.13801/j.cnki.fhclxb.20221223.003.

Author Response

Point 1. Results are mostly described, their detail is not provided, give the reasons and their influence depending on the applied parameters.

Response 1: Thank you for this remark. We have made corresponding changes in the revised manuscript as well as insert the Supplementary Materials Section.

Point 2. Please add quantitative analysis.

Response 2: See Supplementary Materials Information. In this work, we presented printed models, without mechanical analysis. The aim was to show that we can obtain a model for 3D printing based on the CT reconstruction. In the future, it is planned to conduct a thorough examination of the printed models (structures, mechanical properties) that is the goal of separate manuscript.

Point 3. Please state the conclusions of this study.

Response 3: we have insert corresponding subsection into revised manuscript

Point 4. What are the applications of this study? Authors are suggested to provide some technical expressions on the application of the study.

Respond 4: we have represented this information in the Outlook.

Point 5. The author mentions the porous structure in the introduction. But, the introduction section is very briefly organized. The introduction section should be improved using the papers on the subject of this work in such as: doi:10.3390/ma16083262; doi:10.13801/j.cnki.fhclxb.20221223.003.

Response 5: Thank you for this important remark. We have improved the introduction according to your recommendations. See revised manuscript.

Reviewer 2 Report

The paper by Voronkina et al. entitled “Honeycomb Biosilica in Sponges: from understanding of principles of unique hierarchical organization to assessing of biomimetic potential” presents an analysis of the honeycomb-like exoskeleton of a deep-sea glass sponge, followed by the design and fabrication of a bioinspired structure. The authors did a lot of experimental analysis to understand the role of actin filaments within the sponge walls. The manuscript provides information that could be helpful to other researchers. Nevertheless, the following points need to be addressed:

·         The beginning of the introduction should be rewritten. The phrases seem disconnected from each other (Lines 41-50).

·         The introduction finishes talking about future work, why? This is not a proposal.

·         Why is brominated aphrocallistin cited? Does it interfere with the structural changes? Also, if cited, please explain it quickly.

·         Are the images in Figure 1 all from the authors?

·         What is the expected carbon sputtering thickness applied to the samples?

·         Correct “Figure3” to “Figure 3” (Line 231).

·         Figure 14 c-d could be easily made better. For example, use the same decimal point in the numbers, the same font for the numbers, and partially remove the background.

·         The manuscript is often written using the first-person pronoun: “we”. Despite not being a mistake, it decreases the manuscript's formality. It is advisable to modify these with expressions such as “In this paper/study” and so on.

·         The discussion section is hard to read. It starts by advertising a paper from the same research group (???). It is written in a highly informal manner. A drastic rewrite should be made.

Rewrite using the comments stated in the other section.

Author Response

Point 1: The beginning of the introduction should be rewritten. The phrases seem disconnected from each other (Lines 41-50).

Response 1: Thank you for this remark. We have made corresponding changes in the revised manuscript.

Point 2: The introduction finishes talking about future work, why? This is not a proposal.

Response 2: We agreed with your comment and made corresponding changes.    

Point 3: Why is brominated aphrocallistin cited? Does it interfere with the structural changes? Also, if cited, please explain it quickly.

Response 3: We removed this citation from the text. This information is related to metabolic features of this sponge species and not to any kind of structural changes. Numerous species of marine sponges produce such secondary metabolites with mostly antibiotic activities.

Point 4: Are the images in Figure 1 all from the authors?

Response 4: Yes. All images have been made in our Labs.

Point 5: What is the expected carbon sputtering thickness applied to the samples?

Response 5: The expected carbon sputtering thickness applied to the samples is up to 20 nm.

Point 6: Correct “Figure3” to “Figure 3” (Line 231).

Response 6: we have made corresponding correction in the revised manuscript.

Point 7: Figure 14 c-d could be easily made better. For example, use the same decimal point in the numbers, the same font for the numbers, and partially remove the background.

Response 7: Figure 14 is a technical drawing, so we will not remove the background from it. We have changed the drawing and corrected it.

Point 8: The manuscript is often written using the first-person pronoun: “we”. Despite not being a mistake, it decreases the manuscript's formality. It is advisable to modify these with expressions such as “In this paper/study” and so on.

Response 8: We agree with your critical remark and have made corresponding changes in the revised manuscript.

Point 9: The discussion section is hard to read. It starts by advertising a paper from the same research group (???). It is written in a highly informal manner. A drastic rewrite should be made.

Response 9: Thank you very much for this critical remark. We have made corresponding changes in the revised manuscript.We take the liberty to suggest that your remark concerning “hard to read” is related mostly to the fragments within discussion which are related  to structural biology aspects of poriferan glassy skeletons. However, these data are crucial for understanding, especially, for experts in spongology, biomineralization and evolutionary biology who are, definitively, also the readers of this journal. It is difficult to find corresponding balance in the manuscripts submitted to such multidisciplinary journal as Biomimetics.

Round 2

Reviewer 1 Report

The updated version of the manuscript has improved the quality of the presentation of the study. But, reference 6 should be corrected. Delete "(Basel)".

Author Response

Point 1. The updated version of the manuscript has improved the quality of the presentation of the study. But, reference 6 should be corrected. Delete "(Basel)".

Thank you for your valuable remark. We have made corresponding correction in the revised manuscript

Reviewer 2 Report

The authors have made the necessary modifications.

Author Response

Thank you for helping us to improve our manuscript